# Cancer Associated PRDM9: Implications for Linking Genomic Instability and Meiotic Recombination

**DOI:** 10.3390/ijms242216522

**Published:** 2023-11-20

**Authors:** Paris Ladias, Georgios S. Markopoulos, Charilaos Kostoulas, Ioanna Bouba, Sofia Markoula, Ioannis Georgiou

**Affiliations:** 1Laboratory of Medical Genetics in Clinical Practice, Faculty of Medicine, School of Health Sciences, University of Ioannina, 45 110 Ioannina, Greece; parisladias@hotmail.com (P.L.); chkostoulas@gmail.com (C.K.); ibouba@uoi.gr (I.B.); 2Neurosurgical Institute, Faculty of Medicine, School of Health Sciences, University of Ioannina, 45 110 Ioannina, Greece; geomarkop@gmail.com; 3Department of Neurology, Faculty of Medicine, School of Health Sciences, University of Ioannina, 45 110 Ioannina, Greece; smarkoula@uoi.gr

**Keywords:** PRDM9, sequence motif, lung adenocarcinoma, head and neck cancer, breast cancer, ovarian cancer, permutation analysis R, recombination, carcinogenesis

## Abstract

The PR domain-containing 9 or *PRDM9* is a gene recognized for its fundamental role in meiosis, a process essential for forming reproductive cells. Recent findings have implicated alterations in the PRDM9, particularly its zinc finger motifs, in the onset and progression of cancer. This association is manifested through genomic instability and the misregulation of genes critical to cell growth, proliferation, and differentiation. In our comprehensive study, we harnessed advanced bioinformatic mining tools to delve deep into the intricate relationship between *PRDM9F* and cancer. We analyzed 136,752 breakpoints and found an undeniable association between specific PRDM9 motifs and the occurrence of double-strand breaks, a phenomenon evidenced in every cancer profile examined. Utilizing R statistical querying and the Regioner package, 55 unique sequence variations of *PRDM9* were statistically correlated with cancer, from a pool of 1024 variations. A robust analysis using the Enrichr tool revealed prominent associations with various cancer types. Moreover, connections were noted with specific phenotypic conditions and molecular functions, underlining the pervasive influence of *PRDM9* variations in the biological spectrum. The Reactome tool identified 25 significant pathways associated with cancer, offering insights into the mechanistic underpinnings linking *PRDM9* to cancer progression. This detailed analysis not only confirms the pivotal role of *PRDM9* in cancer development, but also unveils a complex network of biological processes influenced by its variations. The insights gained lay a solid foundation for future research aimed at deciphering the mechanistic pathways of *PRDM9*, offering prospects for targeted interventions and innovative therapeutic approaches in cancer management.

## 1. Introduction

*PRDM9* (PR domain-containing 9) is a gene that plays a crucial role in the process of meiosis, which is essential for the formation of reproductive cells [1]. It encodes a protein called PRDM9, which contains a PR/SET domain and zinc finger motifs. These zinc finger motifs allow PRDM9 to bind to specific DNA sequences in the genome. This crucial factor involved in the regulation of genetic recombination and hotspot determination displays a strong affinity for recognizing the specific CCNCCNTNNCCNC motif. This particular DNA sequence pattern plays a fundamental role in guiding the positioning of meiotic recombination hotspots, which is essential for maintaining genetic diversity and fertility. PRDM9’s ability to identify and bind to the CCNCCNTNNCCNC motif is a central element of its function, as it is responsible for targeting DNA double-strand break sites, initiating meiotic recombination events, and, ultimately, contributing to the creation of genetic diversity. This binding interaction highlights the important role of PRDM9 in precisely determining the locations of recombination events and provides insights into the intricate mechanisms by which it orchestrates these crucial genomic processes [2]. Recent studies have uncovered a potential association between PRDM9 sequence motifs and cancer [3]. It has been observed that certain variants of *PRDM9*, particularly those associated with altered zinc finger motifs, may influence the genomic landscape and contribute to the initiation or progression of cancer [4]. In addition, an examination of TCGA data across various cancers disclosed that the *PRDM9* gene exhibited a substantial mutation frequency across numerous cancer types, with mutation rates surpassing 10.0% in certain tumors [5]. Moreover, a subsequent analysis of cancer samples from human patients indicated a pronounced increase in *PRDM9* expression in a multitude of cancer types. In a recent development, non-histone substrates of *PRDM9* have also been detected [6].

As concerns DNA repair and genome stability, *PRDM9* is involved in the regulation of DNA double-strand break repair during meiosis. Dysfunction or variations in *PRDM9* can affect the accuracy of DNA repair mechanisms, leading to genomic instability [4]. Genomic instability is a hallmark of cancer, as it allows for the accumulation of mutations and genetic alterations that promote cancer development [7]. DNA binding specificity is dependent on the zinc finger motifs in the PRDM9 protein, which are responsible for recognizing and binding to specific DNA sequences in the genome. Variations in these motifs can alter the binding specificity of PRDM9, leading to abnormal interactions with DNA. This aberrant DNA binding may result in the misregulation of genes involved in crucial cellular processes, including cell growth, proliferation, and differentiation, while dysregulated gene expression can contribute to oncogenesis [7,8].

*PRDM9* is also involved in epigenetic regulation, as it possesses a PR/SET domain, which is associated with histone methyltransferase activity. Histone modifications, including methylation, play a vital role in epigenetic regulation, influencing nucleosomes and gene expression patterns [9,10]. Altered *PRDM9* variants may disrupt normal histone methylation patterns, affecting the epigenetic landscape and, potentially, leading to the activation or silencing of oncogenes or tumor-suppressor genes [11].

*PRDM9* is a highly polymorphic gene, exhibiting significant genetic variation across individuals [1]. This genetic variability can result in distinct PRDM9 sequence motifs in different individuals, which may have varying affinities for DNA binding and regulatory functions. Certain *PRDM9* variants with specific sequence motifs have been associated with an increased susceptibility to cancer development. A groundbreaking study in 2018 by Houle et al. revealed a surprising observation, as it demonstrated that *PRDM9*, typically restricted to germ cells, is unexpectedly expressed in 20% of cancer tumors, even after rigorous gene homology correction [7]. Furthermore, the significantly elevated expression levels of PRDM9 in tumors, compared to those of healthy neighboring tissues and non-germ tissue databases, indicated its potential significance in cancer biology. The study’s findings also unveiled a strong correlation between aberrant *PRDM9* expression, recurrently mutated regions near *PRDM9* loci, deferentially expressed genes involved in meiotic pathways, and structural variant breakpoints in proximity to PRDM9’s DNA recognition motif, implying an intriguing link between *PRDM9* expression and genomic instability in cancer [7].

It is important to point out that, while the above findings suggested an association between *PRDM9* aberrant expression and cancer, further research is needed to fully understand the underlying mechanisms and establish a definitive causal relationship with PRDM9 sequence motifs variation in somatic cells also. Our previous findings concerning the associations of the common CNVs with PRDM9 consensus sequences [12,13,14], strengthened this hypothesis and presented evidence that the above-mentioned sequences are not randomly distributed throughout the genome and are present in regions where DSBs are occurring, which may indicate a functional mechanism for many types of cancer and their potential diagnostic and prognostic significance.

Here, we describe the correlation between a cancer-specific PRDM9 sequence motif in the vicinity of DNA breaks in four different cancer types—breast invasive carcinoma (BRC), lung adenocarcinoma (LUAD), ovarian serous cystadenocarcinoma (OV), and head and neck squamous cell carcinoma (HN)—from 72 patients from the BreCan database [15].

## 2. Results

There is a little-known association between PRDM9 motifs, key elements of homologous recombination, and cancer. A possible mechanism of double-strand breaks occurrence may be revealed by their presence on both sides from breakpoints in every cancer profile from the 136,752 breakpoints that were tested. For this reason, R statistical querying was used to define the overlap and statistical correlation between our databases [16]. In particular, we demonstrated, using mainly the Regioner package, which variations of the above-mentioned sequence motifs are statistically significant and not just random [17]. The motifs were identified in 100-nucleotide and 500-nucleotide flanking frames, respectively, where the double-strand breaks are the center of this frame. The summary of the datasets that were used is depicted in Table 1.

There were 55 unique sequence variations among the total 1024 PRDM9 sequence variations, resulting in statistically significant genome hits that appeared more frequently in the breakpoint regions with very low *p*-values (in all cases *p* < 0.05). The results are listed in Appendix A. The consensus motifs of all significant PRDM9 motifs are depicted in Figure 1.

In conclusion, we found a significant correlation between the cancer types that were tested with specific motifs of PRDM9 that may mediate recombination. The most exciting finding of this study is the identification of a unique PRDM9 sequence motif, which exhibits the highest statistical significance across all four types of cancer. The overwhelmingly large differences in statistical significance, compared to other statistically significant sequence motifs, suggest that this specific motif d(CCACCATCACCAC) might play a crucial role in the generation of DNA double-strand breaks. Four plots of *p*-values/z-scores corresponding to the specific d(CCACCATCACCAC) sequence motif hits located near to DSBs (500-nt flanking frame) are provided in Figure 2. Respectively, four plots of *p*-values/z-scores corresponding to the d(CCACCATCACCAC) sequence motif hits located to DSBs (100-nt flanking frame) are provided in Figure 3. All z-scores and *p*-values images regarding all sequence motifs that were tested are presented in Appendix A. To bolster the robustness and validity of our findings, we conducted comprehensive ChIP-Seq analysis, as stated in Appendix A. This meticulous approach not only reinforced the authenticity of our results, but also provided crucial mechanistic insights into the molecular underpinnings of our observed effects.

By displacing the PRDM9 motifs outside the sequence boundaries of each individual flanking frame that contains the double-strand breaks, the z-scores drop, demonstrating that the association is dependent on the exact position of the regions, rather than being a non-specific regional effect.

Moreover, an examination was conducted to uncover the dispersion of breakpoints containing the PRDM9 motif in relation to the TSS of genes, utilizing the GREAT tool. The genomic locales of HERV-CREs were identified in proximity to the TSS of 1045 genes (refer to Appendix A). The GREAT evaluation revealed a scattered and gene-centric distribution across the genome.

To elucidate the significant associations of our dataset, we employed the Enrichr tool, an integrative web-based platform for comprehensive gene set enrichment analysis [18]. The gene lists from our dataset were uploaded to the Enrichr platform and using Enrichr’s extensive library of curated gene sets, the data were analyzed to identify significant associations across various categories [18]. The top 10 results across different categories, including diseases, phenotypes, molecular functions, cellular components, biological processes, and transcription factors, were extracted for further interpretation. The Enrichr analysis of our dataset yielded several top-10 results across different categories. In the disease category, the most prominent associations were with kidney cancer, carcinoma, liver cancer, polydactyly, and breast cancer, among others (Jensen Syndrome). Phenotypically, associations were observed with conditions such as Dandy–Walker malformation, short distal phalanx of finger, and abnormality of the humerus (human phenotype ontology). Molecular functions, like ferrous iron binding, G protein-coupled receptor activity, and voltage-gated potassium channel activity, were also highlighted (GO molecular function). Cellular components, such as cell–cell junction, ciliary membrane, and calcium channel complex, were identified (GO cellular component). Furthermore, biological processes, like regulation of cell migration, adenylate cyclase-modulating G protein-coupled receptor signaling pathway, and brain development were significantly enriched (GO biological process). Finally, transcription factors like NFKB1, SOX2, and RUNX2 were found to be associated with specific conditions and cell types [19]. All these significant results are depicted in Figure 3.

Furthermore, to identify the significant pathways associated with cancer, we utilized the Reactome tool, a freely accessible online database of biological pathways [18]. Using the built-in algorithms of Reactome, the data were analyzed to identify significant pathways and the top 25 pathways (Table 2), sorted by *p*-values, were extracted for further analysis. Our analysis revealed the 25 most relevant pathways associated with cancer, sorted by their respective *p*-values (Figure 4). The top pathways include “Regulation of CDH11 Expression and Function” with a *p*-value of 4.60 × 10^−4^, “Regulation of Expression and Function of Type II Classical Cadherins” with a *p*-value of 0.001, and “Regulation of Homotypic Cell-Cell Adhesion”, also with a *p*-value of 0.001. Another noteworthy pathway is the “Ca^2+^ activated K^+^ channels” pathway, with a *p*-value of 0.001. These pathways, among others listed, demonstrate significant associations with cancer development and progression.

Finally, to independently verify our findings, we performed analysis using the Metascape tool [20]. We initially pinpointed all statistically enriched terms that could encompass GO/KEGG terms, based on the Metascape calculation of accumulative hypergeometric *p*-values, along with enrichment factors, and used them as filters. The terms that emerged as significant were subsequently arranged into a hierarchical tree structure based on the kappa-statistical similarities in their gene memberships, a method akin to the one employed by the NCI DAVID site. A kappa score threshold of 0.3 was then applied to segment the tree into distinct clusters of terms. The process yielded a variety of enriched terms across multiple biological processes and pathways, including the establishment of cell polarity (GO:0030010), signaling by GPCR (R-HSA-372790), a smoothened signaling pathway (GO:0007224), and an enzyme-linked receptor protein signaling pathway (GO:0007167). Further notable terms were extracellular matrix organization (R-HSA-1474244), regulation of growth (GO:0040008), regulation of membrane potential (GO:0042391), regulation of hydrolase activity (GO:0051336), and regulation of monoatomic ion transport (GO:0043269). Additional areas such as the neuronal system (R-HSA-112316), regulation of neuron projection development (GO:0010975), modulation of chemical synaptic transmission (GO:0050804), adenylate cyclase-modulating G protein-coupled receptor signaling pathway (GO:0007188), metal ion transport (GO:0030001), cell morphogenesis (GO:0000902), heart development (GO:0007507), skeletal system development (GO:0001501), tube morphogenesis (GO:0035239), and embryonic morphogenesis (GO:0048598) were also represented (Figure 5).

## 3. Discussion

The results of our study provide irrefutable evidence of a significant association between specific PRDM9 sequence motifs and various cancer types, shedding light on the potential role of *PRDM9* in cancer development and progression.

One of the key findings of this study is the identification of a unique PRDM9 sequence motif, represented as d(CCACCATCACCAC), which exhibits an exceptionally strong statistical significance across all four cancer types studied (breast invasive carcinoma, lung adenocarcinoma, ovarian cerous Cystadenocarcinoma, and head and neck squamous cell carcinoma). This motif consistently appears near DNA double-strand break (DSB) sites, suggesting its involvement in the generation of DSBs. The remarkable difference in statistical significance, compared to other motifs, underscores its potential significance in cancer biology. The latest bibliographical evidence supports such a connection [21,22]. Importantly, a recent study by Houle et al. revealed that *PRDM9*, typically expressed solely in germ cells, was unexpectedly present in 20% of 1879 cancer samples across 39 cancer types, with its expression levels significantly higher in tumors than in healthy tissues. This aberrant expression correlates with genomic instability in cancer, linking the meiosis-specific gene to cancer, in total accordance with the results presented in this study and, under the same conceptual framework, could combinatorially provide a functional role of *PRDM9* in carcinogenesis [7]. Our focus has been on a comprehensive bioinformatics approach, and we believe our findings can serve as a groundwork for future studies that include experimental validations. However, CHIP-seq experiments in future studies are needed to directly demonstrate the binding of PRDM9 to the identified motifs to further support the findings.

The presence of this specific PRDM9 motif near DSBs raises questions about its role in genomic instability, a hallmark of cancer. Genomic instability, resulting from faulty DNA repair mechanisms, facilitates the accumulation of mutations and genetic alterations that contribute to cancer development. The strong association between the identified PRDM9 motif and DSBs suggests a mechanistic link between *PRDM9* and the initiation or progression of cancer through genomic instability. Our study also highlights the genetic variability of *PRDM9* across individuals, leading to distinct sequence motifs. Certain *PRDM9* variants exhibit statistically significant associations with cancer types, emphasizing the importance of considering individual genetic profiles in cancer-risk assessment and personalized medicine. Tailoring prevention and treatment strategies based on *PRDM9* variants could enhance the effectiveness of cancer management. In line with this, Kaiser et al. revealed that chromatin loop anchor points, crucial for nuclear organization, are characterized by unique patterns of somatic and germline variations, in part in PRDM9 motifs, and exhibit elevated rates of variations, indicating regulatory functions in genomic instability in cancer that are associated with genetic variations [23].

The integration of Enrichr and Reactome analyses further supports the relevance of *PRDM9* in cancer biology. Enrichr analysis reveals associations between *PRDM9*-related data and various categories, including diseases, molecular functions, cellular components, biological processes, and transcription factors, with notable connections to cancer and cancer-related pathways. The Reactome analysis identifies relevant pathways such as “Regulation of CDH11 Expression and Function” and “Regulation of Homotypic Cell-Cell Adhesion [24]“, reinforcing the significance of *PRDM9* in cancer development and progression. Importantly, tumor-suppressor factors such as CDH1 ansd CDH11 are associated with genomic stability [25,26]. Their interplay with *PRDM9* might represent a novel avenue that connects genome integrity to carcinogenesis. In addition, cell adhesion is a critical factor that regulates the progression of cancer [24]. The possible involvement of *PRDM9* has not been previously reported and is also an interesting avenue for therapeutic interventions. The use of the Metascape tool further confirmed the aforementioned findings. Several of the identified pathways and gene sets are critically involved in tumorigenesis and cancer progression. The establishment of cell polarity (GO:0030010) is a fundamental process that, when disrupted, has been implicated in the loss of tissue organization characteristic of cancerous tissues [27]. The signaling by G protein-coupled receptors (GPCRs) (R-HSA-372790), which includes the smoothened signaling pathway (GO:0007224), plays a notable role in cell proliferation and survival, pathways often hijacked in cancer to promote growth and evade apoptosis [28].

Enzyme-linked receptor protein signaling (GO:0007167) and the regulation of growth (GO:0040008) are also crucial, as they can be linked to the overactive signaling cascades frequently observed in cancer cells, leading to uncontrolled proliferation. Similarly, extracellular matrix organization (R-HSA-1474244) is a key factor in cancer metastasis, where alterations in the extracellular matrix can aid tumor cells in invasion and dissemination throughout the body. The regulation of hydrolase activity (GO:0051336), metal ion transport (GO:0030001), and various developmental processes, such as heart (GO:0007507) and skeletal system development (GO:0001501), underscore the complex interplay between cellular differentiation pathways and cancer, where the reactivation of developmental programs can lead to the formation of tumors with stem-cell-like features and a high degree of plasticity [28].

The involvement of the neuronal system (R-HSA-112316) highlights the emerging field of cancer neuroscience, which examines the interplay between neural processes and cancer development, particularly in terms of how tumors can coopt neuronal signaling for growth and survival. Furthermore, the modulation of synaptic transmission (GO:0050804) can be reflective of the recently acknowledged phenomenon of synaptic-like connections between cancer cells, contributing to tumor progression [29].

The interconnectivity highlighted by the network analysis underscores the complex nature of cancer biology. In essence, the network of enriched terms forms a multi-faceted landscape that reflects the complex biological underpinnings of tumorigenesis, suggesting that PRDM9-associated genomic regions may be involved in a novel mechanism that contributes to this process. In addition, the enriched pathways might also present potential targets for therapeutic intervention.

Overall, the presented findings have several clinical implications. First, the specific PRDM9 motif identified in this study could serve as a potential diagnostic marker for cancer, allowing for early detection and more accurate risk assessment. Second, understanding the mechanistic role of this motif in DSB generation may lead to the development of targeted therapies aimed at disrupting this process in cancer cells.

In conclusion, our results provide strong evidence of a significant correlation between specific PRDM9 sequence motifs and multiple cancer types. The identification of a unique, highly significant PRDM9 motif suggests a potential mechanistic link between *PRDM9*, genomic instability, and cancer. This study opens promising avenues for further research into the role of *PRDM9* in cancer biology and the development of novel diagnostic and therapeutic strategies.

## 4. Materials and Methods

### 4.1. Bioinformatic Mining Tools for the Identification of PRDM9 Residing within the Boundaries of a Flanking Region that Contains Cancer Breakpoints

For the implementation of this study, we initially collected the dataset from the online database BreCAN [12]. This specific database contains stored profiles of cancer patients whose genome has undergone focal amplifications and deletions. More specifically, the profiles deposited on the database are genomes of somatic cells with mapped double-stranded breaks from each cancer profile. The 72 patients studied suffered from 4 different types of cancer—15 with breast invasive carcinoma (BRC), 17 with lung adenocarcinoma (LUAD), 23 with ovarian serous cystadenocarcinoma (OV), and 17 with head and neck squamous cell carcinoma (HN). The analysis was conducted overall in 136.752 breakpoints [12].

The profiles of the patients’ genomes for breast cancer (BRC) were obtained from the analysis of the association of various breast cancer subtypes, performed within experiments by the Banerji S group using the dbGaP database (database of genotypes and phenotypes) [30]. In contrast, the data for the other types of cancer were obtained from The Cancer Genome Atlas (TCGA) database. The association of the patients’ genomes was performed according to version hg19 of the genome, while the meerkat algorithm was used for mapping the entire genome of each patient at the nucleotide level, maintaining the default analysis parameters [31]. The double-stranded breaks detected through high-coverage next-generation sequencing reflected the structural genetic rearrangements of the genome and are included overall.

As a further step in analyzing the relationship between PRDM9 motifs and sequence data, we created four databases in BED format associated with the specific four cancer types across the human genome. Each bed file contains the coordinates of a 100- and 500-nucleotide flanking frame, respectively, where every observed double-strand break is at the center of these frames. Correspondingly, we also created a database in BED format relating to the PRDM9-related sequences from the degenerated sequence motif (CCNCCNTNNCCNC).

Below are the algorithmic steps that we followed:

1. R 3.5.1 (http://www.r-project.org/, accessed on 2 June 2023) [16] was used to perform the permutation analysis and the overlap analysis between PRDM9 hits and breakpoint regions. Based on the permutation tests, we evaluated the statistical associations between our loci sets using the Regioner package, which was developed specifically for this purpose [17]. To extend the functionality of the permutation test framework, the user can provide custom functions. There are several predefined randomization and evaluation functions included in this framework that are designed specifically for working with genomic regions. The exact algorithm and scripts are listed in the Appendix A. With the Regioner package, we evaluated the associations between our region gene sequence sets based on the permutation tests. Below is a description of the methodology used to detect sequence variations of PPRDM9 motifs that are statistically correlated with the related types of cancer.

2. Data collection was implemented by utilizing a script written in C# that identified all different forms of PRDM9 sequence motifs (CCNCCNTNNCCNC). A total of 1024 different PRDM9 sequence variations were identified.

3. R’s matchPattern function was applied to all chromosomes of the genome. As a result, it produced a text file with all hits containing PRDM9. Second, R’s regioneR library was used to perform the overlapPermTest function [17]. Next, R’s regioneR library was used to perform R’s local ZScore function. In the next step, overlap PermTest plots and local ZScore plots were saved as JPG images. Following that, the overlap region’s function in R was performed using R’s regioneR library. In the last step, the output of the overlap region’s function was stored in a text file. A *p*-value < 0.05 and a Z-score < −3 or >3 were considered significant. Additionally, the UCSC genome browser provides bioinformatic tools [32] to manually screen the human genome for PRDM9 sequence motifs residing within the boundaries of breakpoint regions and their surrounding sequences in each of the 24 human chromosomes. The default settings were used as the gold standard in our analyses.

### 4.2. Genomic Annotation and Analysis of Genomic Distribution

Genomic annotation and analysis were performed, following previously established systems biology approaches [33,34,35]. Gene-specific analysis was performed with the Genomic Regions Enrichment of Annotations Tool (GREAT) (great.stanford.edu, accessed on 2 September 2023) [36]. GREAT was used to analyze the relative distribution between cancer-breakpoint loci and transcription start sites (TSSs) of human genes, using the default association rule settings for statistical analysis, following established protocols [37]. The cancer-associated genes were downloaded from the GREAT server and visualized in the UCSC genome browser. To elucidate the significant associations of our dataset, we employed the Enrichr tool [18], an integrative web-based platform for comprehensive gene set enrichment analysis (https://maayanlab.cloud/Enrichr/, accessed on 2 September 2023). The gene lists from our dataset were uploaded to the Enrichr platform and, using Enrichr’s extensive library of curated gene sets, the data were analyzed to identify significant associations across various categories. The top 10 results across different categories, including diseases, phenotypes, molecular functions, cellular components, biological processes, and transcription factors, were extracted for further interpretation. Finally, to identify the significant pathways associated with cancer, we utilized the Reactome tool, a freely accessible online database of biological pathways (https://reactome.org, accessed on 2 September 2023). Using the built-in algorithms of Reactome, the data were analyzed to identify significant pathways and the top 25 pathways, sorted by *p*-values, were extracted for further analysis following the GREAT analysis, as described elsewhere [38]. Gene lists of interest were also subjected to enrichment analysis, using the Metascape bioinformatics tool (http://metascape.org, accessed on 1 November 2023) [38].

### 4.3. Enrichment Analysis with Chip-Seq Data

The enrichment analysis with ChIP-seq data from ChIP-Atlas (https://chip-atlas.org/last, accessed on 25 October 2023), was performed to uncover regulatory elements and biological insights associated with specific protein-DNA interactions. ChIP-Atlas is a comprehensive repository of ChIP-seq datasets that catalogues experimental results, including transcription factor binding profiles, histone modifications, and other chromatin-associated events across various cell types and conditions. For our analysis, we included our customized databases with all DSBs in 4 types of cancer with the respected degenerate PRDM9 sequence motif (CNNCNNTNNCNC). These datasets were processed and aligned to the reference genome using the Bowtie2 and MACS2 algorithms. Peak calling was conducted to identify genomic regions enriched with specific protein binding events, and these peaks were subsequently used for downstream analyses, including motif evaluation.

## 5. Conclusions

Our comprehensive investigation into *PRDM9* unveiled its multifaceted role in cancer initiation and progression. The specificity of PRDM9’s zinc finger motifs in DNA binding underscores the gene’s crucial contribution to genomic stability and gene regulation. With 55 unique PRDM9 sequence variations linked to a myriad of cancers, the gene’s influence is expansive, echoing through various phenotypic conditions and molecular functions. Our analysis has illuminated the intersections between *PRDM9* variations and intricate biological pathways, underlining the gene’s pervasive impact. The prominence of associations with cancers underscores the urgency and necessity of in-depth studies geared toward deciphering the underlying mechanisms. The identified pathways, such as regulation of CDH11 and Type II classical cadherins, might serve as focal points for future targeted cancer research. Considering these findings, the interplay between PRDM9′s sequence motifs and the genomic landscape emerges as a critical frontier in cancer research. The robust correlations that were identified not only reinforce the gene’s pivotal role, but also open avenues for targeted therapeutic strategies. A deeper exploration into *PRDM9*’s role in genomic instability and abnormal gene expression is now imperative. This exploration holds the promise for more personalized, targeted, and effective therapeutic interventions, marking a significant stride toward cancer eradication.

## Figures and Tables

**Figure 1 ijms-24-16522-f001:**
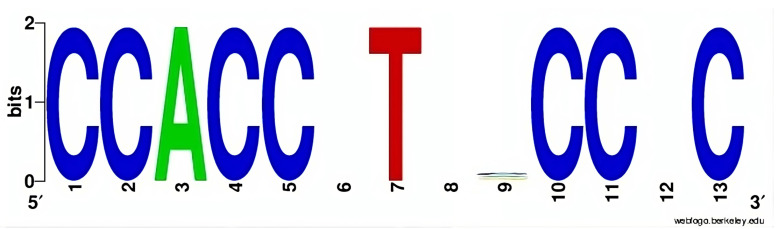
Consensus sequence motif of all 55 unique sequence variations of the total 1024 PRDM9 sequence variations that are correlated significantly with the four types of cancers in all of the 136,752 cancer breakpoints.

**Figure 2 ijms-24-16522-f002:**
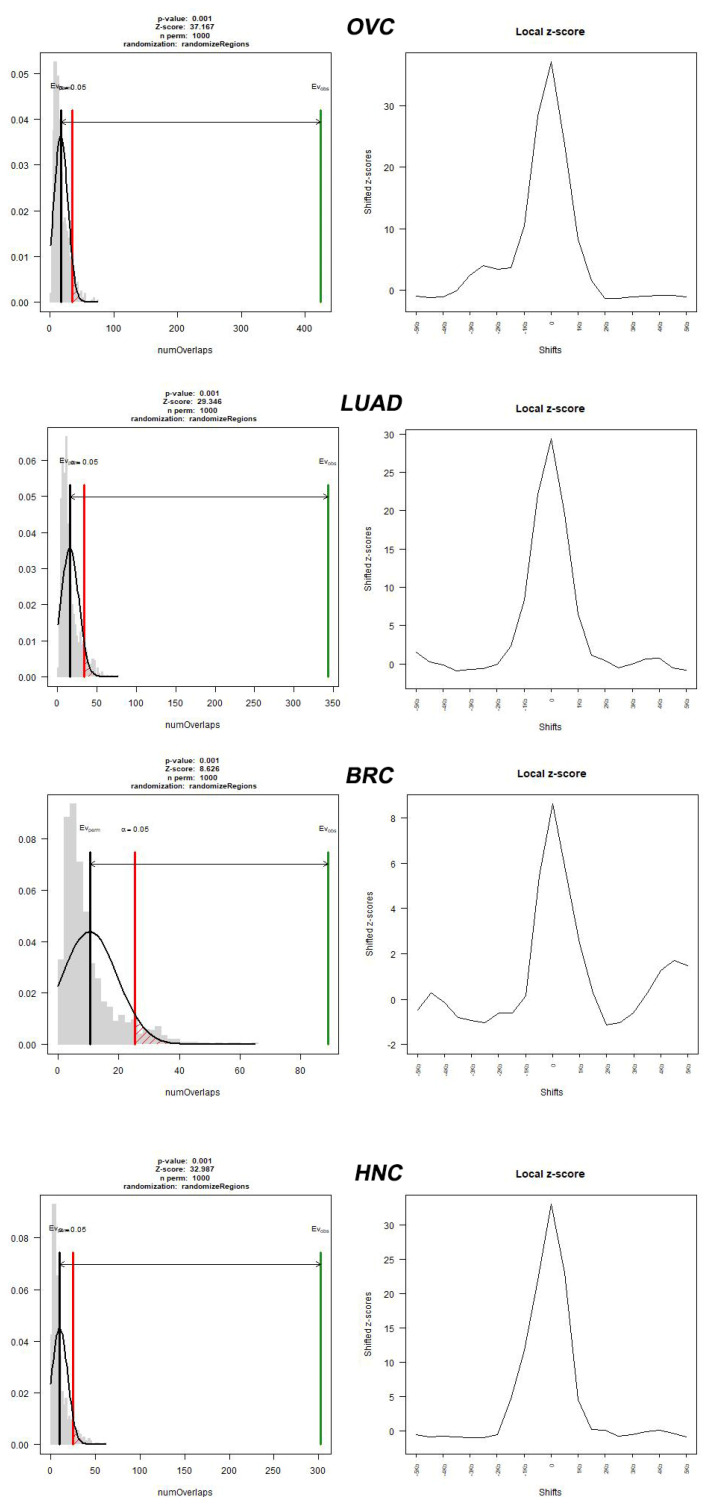
Breakpoints and d(CCACCATCACCAC) overlap permutation test for various type of cancer of the respective flanking frame of 500-nt. The Figure 2 depicts a gray histogram representing the evaluation of the randomized region set with a fitted normal and a black bar representing the mean of the randomized evaluations. Accordingly, the green line represents the observed hits (Evobs), while the red line represents the expected hits (Evexp), and shows the limit at which the correlation between d(CCACCATCACCAC) and 4 types of cancer can be statistically or by chance associated. The *p*-value is equal to 0.05 when the green line coincides with the red line. The results of the test include ovarian cancer (OVC), with *p*-value = 0.0014 and Z-score = 37.167; lung adenocarcinoma (LUAD), with *p*-value = 0.001 and Z-score = 29.346; breast cancer, with *p* value = 0.001 and Z-score = 8.628; and head and neck cancer (HNC), with *p*-value = 0.001 and Z-score = 32.987.

**Figure 3 ijms-24-16522-f003:**
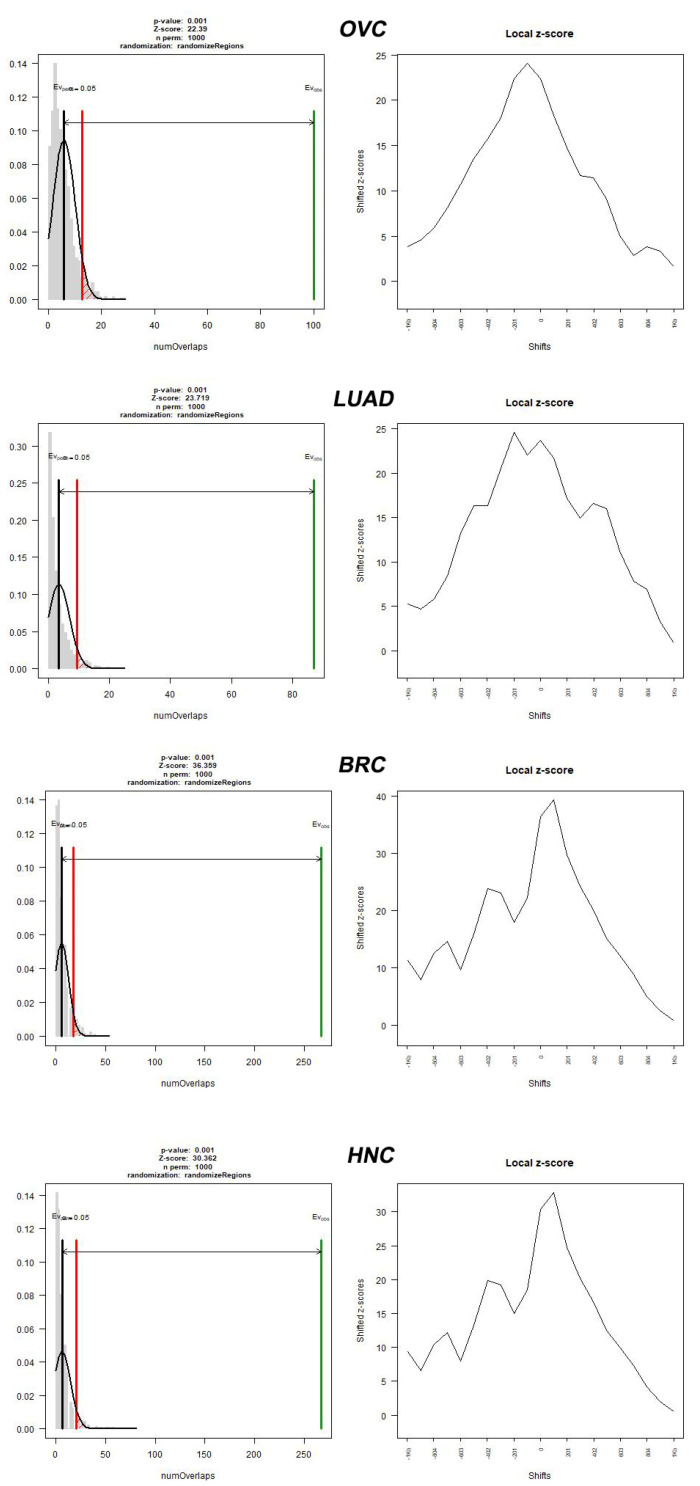
Breakpoints and d(CCACCATCACCAC) overlap permutation test for various type of cancer of the respective flanking frame of 100 nt. The Figure 3 depicts a gray histogram representing the evaluation of the randomized region set with a fitted normal and a black bar representing the mean of the randomized evaluations. Accordingly, the green line represents the observed hits (Εvobs), while the red line represents the expected hits (Evexp), and shows the limit at which the correlation between d(CCACCATCACCAC) and the 4 types of cancer can be statistically or by chance associated. The *p*-value is equal to 0.05 when the green line coincides with the red line The results of the test include ovarian cancer (OVC), with *p*-value = 0.001 and Z-score = 22.39; lung adenocarcinoma (LUAD), with *p*-value = 0.001 and Z-score = 23.719; breast cancer, with *p* value = 0.001 and Z-score = 36.359; and head and neck cancer (HNC), with *p*-value = 0.001 and Z-score = 30.362.

**Figure 4 ijms-24-16522-f004:**
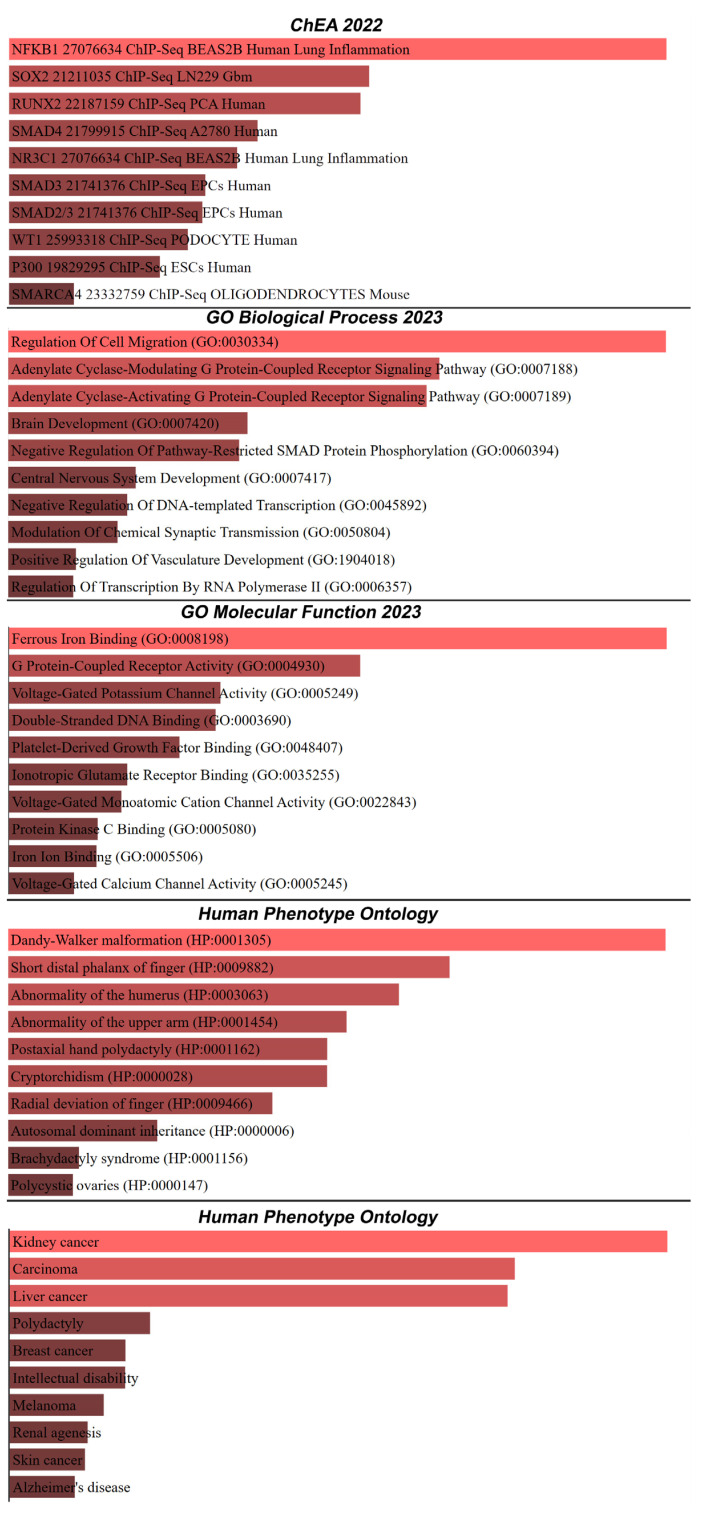
Results of comprehensive gene set enrichment analysis, including diseases, phenotypes, molecular functions, cellular components, biological processes, and transcription factors.

**Figure 5 ijms-24-16522-f005:**
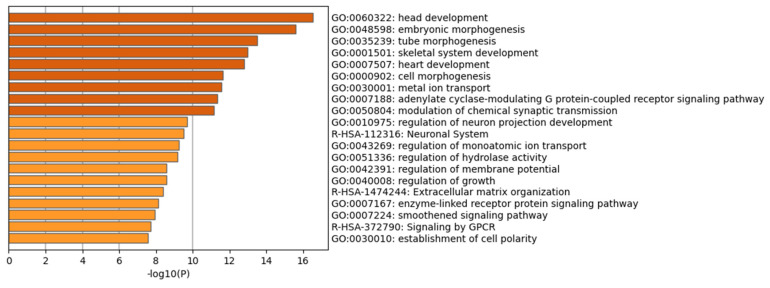
Bar graph of enriched terms across input gene lists, colored by *p*-values, based on Metascape analysis.

**Table 1 ijms-24-16522-t001:** Summary of datasets used.

Cancer Type	Source	Number of Cancer-Normal Pairs	Number of Breakpoints	Number of Breakpoints per Cancer Genome
Breast invasive carcinoma (BRC)	Banerji et al.	15 pairs	22,077	1471
Lung adenocarcinoma (LUAD)	TCGA	17 pairs	36,086	2122
Ovarian serous cystadenocarcinoma (OV)	TCGA	23 pairs	56,435	2453
Head and neck squamous cell carcinoma (HN)	TCGA	17 pairs	22,154	1303
**Total**		**72 pairs**	**136,752**	**7349**

**Table 2 ijms-24-16522-t002:** Top 25 enriched pathways associated with PRDM9, according to Reactome analysis.

Pathway Name	Found	Total	*p* Value	FDR	Found	Ratio
Regulation of CDH11 Expression and Function	10/35	2 × 10^−3^	4 × 10^−4^	4.69 × 10^−3^	25/28	2 × 10^−3^
Regulation of Expression and Function of Type II Classical Cadherins	10/39	3 × 10^−3^	1 × 10^−3^	4.69 × 10^−3^	26/33	2 × 10^−3^
Regulation of Homotypic Cell-Cell Adhesion	10/39	3 × 10^−3^	1 × 10^−3^	4.69 × 10^−3^	26/33	2 × 10^−3^
Ca^2+^ activated K^+^ channels	5/10	6.57 × 10^−4^	1 × 10^−3^	4.69 × 10^−3^	2/3	2.10 × 10^−4^
NOTCH2 intracellular domain regulates transcription	6/16	1 × 10^−3^	2 × 10^−3^	5.35 × 10^−3^	9/9	2.10 × 10^−4^
Adherens junctions interactions	13/66	4 × 10^−3^	2 × 10^−3^	5.63 × 10^−3^	31/49	6.99 × 10^−4^
Regulation of CDH11 function	5/13	8.53 × 10^−4^	4 × 10^−3^	5.63 × 10^−3^	10/10	3.49 × 10^−4^
Formation of intermediate mesoderm	5/13	8.53 × 10^−4^	4 × 10^−3^	5.63 × 10^−3^	4/5	9.79 × 10^−4^
Formation of lateral plate mesoderm	4/8	5.25 × 10^−4^	4 × 10^−3^	5.63 × 10^−3^	4/5	6.29 × 10^−4^
HS-GAG biosynthesis	9/39	3 × 10^−3^	4 × 10^−3^	5.63 × 10^−3^	13/14	2.80 × 10^−4^
Other semaphorin interactions	6/19	1 × 10^−3^	4 × 10^−3^	5.63 × 10^−3^	5/9	3.49 × 10^−4^
Regulation of CDH11 mRNA translation by microRNAs	5/14	9.19 × 10^−4^	5 × 10^−3^	6.53 × 10^−3^	4/4	9.79 × 10^−4^
RUNX3 regulates WNT signaling	4/10	6.57 × 10^−4^	8 × 10^−3^	7.61 × 10^−3^	5/5	6.29 × 10^−4^
MECP2 regulates transcription factors	4/10	6.57 × 10^−4^	8 × 10^−3^	7.61 × 10^−3^	4/8	2.80 × 10^−4^
Defective EXT1 causes exostoses 1, TRPS2 and CHDS	5/16	1 × 10^−3^	8 × 10^−3^	7.61 × 10^−3^	4/4	3.49 × 10^−4^
Defective EXT2 causes exostoses	5/16	1 × 10^−3^	8 × 10^−3^	7.61 × 10^−3^	4/4	5.59 × 10^−4^
Gastrulation	20/143	9 × 10^−3^	8 × 10^−3^	7.61 × 10^−3^	43/72	2.80 × 10^−4^
Signaling by NOTCH2	8/38	2 × 10^−3^	1 × 10^−2^	7.61 × 10^−3^	14/20	2.80 × 10^−4^
Cristae formation	7/31	2 × 10^−3^	1.1 × 10^−2^	7.61 × 10^−3^	2/2	5 × 10^−3^
Germ layer formation at gastrulation	6/24	2 × 10^−3^	1.1 × 10^−2^	7.61 × 10^−3^	11/11	1 × 10^−3^
RUNX3 regulates BCL2L11 (BIM) transcription	3/6	2 × 10^−3^	1.1 × 10^−2^	7.61 × 10^−3^	2/2	1.40 × 10^−4^
Vasopressin-like receptors	3/6	3.94 × 10^−4^	1.1 × 10^−2^	7.61 × 10^−3^	5/7	4.89 × 10^−4^
cGMP effects	5/18	1 × 10^−3^	1.3 × 10^−2^	7.61 × 10^−3^	2/4	2.80 × 10^−4^
Extracellular matrix organization	37/328	2.2 × 10^−3^	1.4 × 10^−2^	7.61 × 10^−3^	150/319	2.20 × 10^−4^
Regulation of CDH11 gene transcription	4/12	7.88 × 10^−4^	1.4 × 10^−2^	7.61 × 10^−3^	11/14	9.79 × 10^−4^

## Data Availability

Data are contained within the article and Appendix A.

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
