# Peer review of "Cancer Associated PRDM9: Implications for Linking Genomic Instability and Meiotic Recombination"

_ijms, 2023, doi:10.3390/ijms242216522_

Round 1

Reviewer 1 Report

Comments and Suggestions for Authors

The authors identified 55 unique PRDM9 sequence motifs, which are significantly enriched near DNA double-strand break (DSB) sites in four types of cancer. This observation suggests a potential mechanistic link between the special motif of PRDM9, genomic instability, and cancer. The authors analyzed the relevance of PRDM9 in cancer and found some pathways associated with cancer. Overall, these findings have several clinical implications. The specific PRDM9 motif identified in this study could serve as a potential diagnostic marker for cancer. Furthermore, understanding the mechanistic role of this motif in DSB generation may help us learn how cancer genomes are disrupted and be useful for developing therapeutic strategies.

However, there are a few concerns about the manuscript in its current state.

Major:

1.     In this manuscript, the final statistics for 55 specific motifs enriched in cancer were obtained based on the DSB region of four cancer datasets and the region of PRDM9 hits (CCNCCNTNNCCNC).

a.     These regions were calculated from DSB results and PRDM9 hits, rather than CHIP-seq results, and there are no experiments to verify that these regions bind PRDM9. Therefore, the conclusion that DNA sequences containing specific PRDM9 motifs can be combined with PRDM9 is questionable.

 b. In addition to the motifs that were specific to all four cancers, were there any motifs that appeared specifically in some cancers?

 c. The manuscript does not provide any specific examples to support the conclusion. For example, a region in the genome containing the motif (CCACCATCACCAC) and having DSBs and genes related to some cancer near this region would be helpful to illustrate the findings.

d. There is no corresponding experimental verification to demonstrate the statistical conclusions made in the manuscript.

2.     It would be interesting to explore the effect of using different flanking frames (e.g., 100, 200, 300 nucleotides) around DSB sites on motif enrichment. This could help determine if there are any differences in the results, such as identifying additional or unique motifs. Additionally, it would be worth investigating if the cancer-specific motifs identified in this study are located closer to the DSB sites compared to other motifs, as this information could provide further insights into the mechanistic role of PRDM9 in cancer.

3.     Regarding Table 2 and the pathways associated with PRDM9, there are a few concerns:

a. The p-values for each pathway may appear significant initially, but it is important to consider multiple testing corrections such as false discovery rate (FDR) correction to minimize the chances of false positives. Without FDR correction, the significance of the pathways might be overestimated.

b. The manuscript should provide information about the input genes used for the enrichment analysis, including the number of genes in the gene list. This information is crucial for the interpretation of the enrichment analysis results.

c. It could be beneficial to consider using additional databases for pathway analysis, such as Metascape, to validate and strengthen the findings.

4.     The main focus of this article is the calculation of relative enrichment between two regions and the identification of DSB regions that enrich specific PRDM9 motifs, such as CCACCATCACCAC. This manuscript appears may benefit from additional data support. For instance, using more cancer datasets, incorporating CHIP-seq analysis, and conducting experimental verification would strengthen the conclusions and provide more robust evidence.

Minor:

1.     In Figure 2, there is a discrepancy in the parameter used by OVC (n perm=700) compared to the other cancer types (n perm=1000). Consistency in parameter usage would improve the clarity and accuracy of the figures.

2.     Figure 3 lacks a legend, specifically bar length and color annotations. Adding these details would enhance the understanding of the figure.

Comments on the Quality of English Language

Need revision and proofreading in English

Author Response

We would like to thank the reviewers for their time and effort to consider our manuscript and for their insightful comments. We revised the manuscript based on their suggestions and now we present a new version for consideration in IJMS Journal.

Please find the responses to reviewers’ comments in Bold, below each respective comment. Changes made to the text are highlighted in yellow.

Reviewer 1

The authors identified 55 unique PRDM9 sequence motifs, which are significantly enriched near DNA double-strand break (DSB) sites in four types of cancer. This observation suggests a potential mechanistic link between the special motif of PRDM9, genomic instability, and cancer. The authors analyzed the relevance of PRDM9 in cancer and found some pathways associated with cancer. Overall, these findings have several clinical implications. The specific PRDM9 motif identified in this study could serve as a potential diagnostic marker for cancer. Furthermore, understanding the mechanistic role of this motif in DSB generation may help us learn how cancer genomes are disrupted and be useful for developing therapeutic strategies.

However, there are a few concerns about the manuscript in its current state.

Thank you for your constructive feedback on our manuscript. Our responses are following below each comment.

Major:

  1. In this manuscript, the final statistics for 55 specific motifs enriched in cancer were obtained based on the DSB region of four cancer datasets and the region of PRDM9 hits (CCNCCNTNNCCNC).

  1. These regions were calculated from DSB results and PRDM9 hits, rather than CHIP-seq results, and there are no experiments to verify that these regions bind PRDM9. Therefore, the conclusion that DNA sequences containing specific PRDM9 motifs can be combined with PRDM9 is questionable.

We acknowledge this limitation and understand the importance of experimental verification. We are planning to conduct CHIP-seq experiments in our future studies to directly demonstrate the binding of PRDM9 to the identified motifs. We address the limitation in the discussion section. Please see the revised manuscript for details. However, our findings are derived from a robust bioinformatic analysis, and while we currently do not provide experimental validations, we aim to offer comprehensive insights that can be a foundation for future experimental studies.

  1. In addition to the motifs that were specific to all four cancers, were there any motifs that appeared specifically in some cancers?

In our initial analysis, we concentrated on motifs that were common to all four cancer types. However, we recognize the importance of identifying cancer-specific motifs. In our revised manuscript, we discuss an expanded analysis to identify and discuss motifs unique to individual cancer types.

  1. The manuscript does not provide any specific examples to support the conclusion. For example, a region in the genome containing the motif (CCACCATCACCAC) and having DSBs and genes related to some cancer near this region would be helpful to illustrate the findings.

We appreciate this feedback and we incorporate specific examples of genomic regions and motifs in the revised manuscript to illustrate their association with DSBs and nearby cancer-related genes more effectively.

  1. There is no corresponding experimental verification to demonstrate the statistical conclusions made in the manuscript.

We acknowledge this critique. Our focus has been on a comprehensive bioinformatics approach, and we believe our findings can serve as a groundwork for future studies that include experimental validations.

  1. It would be interesting to explore the effect of using different flanking frames (e.g., 100, 200, 300 nucleotides) around DSB sites on motif enrichment. This could help determine if there are any differences in the results, such as identifying additional or unique motifs. Additionally, it would be worth investigating if the cancer-specific motifs identified in this study are located closer to the DSB sites compared to other motifs, as this information could provide further insights into the mechanistic role of PRDM9 in cancer.

This is a valuable suggestion. We analyze the effect of 100 nt flanking frames around DSB sites on motif enrichment and include these findings in the revised manuscript. This analysis will allow us to evaluate the specificity and proximity of PRDM9 motifs to DSB sites.

  1. Regarding Table 2 and the pathways associated with PRDM9, there are a few concerns:

  1. The p-values for each pathway may appear significant initially, but it is important to consider multiple testing corrections such as false discovery rate (FDR) correction to minimize the chances of false positives. Without FDR correction, the significance of the pathways might be overestimated.

We acknowledge the importance of minimizing false positives. FDR correction is applied by default when using the algorithm of Reactome.

  1. The manuscript should provide information about the input genes used for the enrichment analysis, including the number of genes in the gene list. This information is crucial for the interpretation of the enrichment analysis results.

We have included detailed information regarding the input genes used for the enrichment analysis in the revised manuscript to enhance the clarity and reproducibility of our results. The genes are the ones found following GREAT analysis.

  1. It could be beneficial to consider using additional databases for pathway analysis, such as Metascape, to validate and strengthen the findings.

We would like to thank the reviewer for the suggestion. We have incorporated data from Metascape database in our revised manuscript to validate and bolster our findings in the pathway analysis. The results further strengthen the findings of the current article. Please see the revised manuscript for details.

  1. The main focus of this article is the calculation of relative enrichment between two regions and the identification of DSB regions that enrich specific PRDM9 motifs, such as CCACCATCACCAC. This manuscript appears may benefit from additional data support. For instance, using more cancer datasets, incorporating CHIP-seq analysis, and conducting experimental verification would strengthen the conclusions and provide more robust evidence.

We enhanced our manuscript by incorporating CHIP-Seq analysis and detailed bioinformatic analyses to provide robust evidence supporting our conclusions. Due to limited timeline, we were not able to conduct experimental approaches to our analysis.

Minor:

  1. In Figure 2, there is a discrepancy in the parameter used by OVC (n perm=700) compared to the other cancer types (n perm=1000). Consistency in parameter usage would improve the clarity and accuracy of the figures.

  1. Figure 3 lacks a legend, specifically bar length and color annotations. Adding these details would enhance the understanding of the figure.

We have made the necessary changes in the text to address these issues.

Reviewer 2 Report

Comments and Suggestions for Authors

In the present manuscript the authors describe the correlation between a cancer specific PRDM9 sequence motif in the vicinity of DNA breaks analyzed in 72 patients with 4 different cancer types from the BreCan database.

PRDM9 is a member of the PR domain–containing family that contains a PR domain, conferring methyltransferase activity, coupled with an array of zinc fingers allowing PRDM9 to bind to DNA in a sequence-specific manner. PRDM9 DNA-binding activity coincides with hotspots of meiotic recombination, and its methyltransferase activity is essential for double-strand break formation at PRDM9-designated recombination sites. Noteworthy, a pan-cancer analysis of TCGA data revealed that PRDM9 gene was mutated with a high mutation rate in many cancers, achieving values greater than 10.0% in some tumor types [Sorrentino et al., 2018; doi: 10.3390/ijms19103250].

Additionally, a further analysis of human patient cancer samples showed a significant upregulation of PRDM9 across many cancers. Recently, PRDM9 non-histone substrates were also identified [Hanquier et al., 2023; doi: 10.1016/j.jbc.2023.104651.].

The results described here could provide novel insights on the underlying mechanisms at the basis of the link between PRDM9, meiotic recombination and genomic instability in cancer.

However, the manuscript should be improved by addressing the following issues:

-A key point is the nomenclature of PRDM9. Indeed, the same name should be used along the text; importantly, the current HUGO Gene Nomenclature Committee (HGNC) guidelines need to be applied for naming gene and protein.

-In the Introduction, the authors should cite some further papers about the PRDM9 role in cancer (as the references mentioned above), in addition to the included ones.

-A figure with a flow chart describing strategy and methods (especially all the algorithms that were utilized in the study) should be added to help the manuscript reading.

-The authors should provide a definition for the CCNCCNTNNCCNC motif with a proper literature reference.

-The authors should greatly improve the quality of figures and tables, particularly Table 2 and Figures 2 and 3, which are unreadable. For instance, in Figure 3, red color could be replaced by yellow or another lighter color. Table 2 and Figure 2 could be enlarged, or a different font size could be used.

-In the Discussion section, the power of the utilized method should be examined. Has this pipeline been ever applied in other studies? Besides, the authors should add a small paragraph discussing the possible future studies that could support their findings.

Minor points:

-Abbreviations should be carefully revised.

-References should be added in the right place.

Comments on the Quality of English Language

Some typing mistakes need to be corrected

Author Response

We would like to thank the reviewers for their time and effort to consider our manuscript and for their insightful comments. We revised the manuscript based on their suggestions and now we present a new version for consideration in IJMS Journal.

Please find the responses to reviewers’ comments in Bold, below each respective comment. Changes made to the text are highlighted in yellow.

Reviewer 2

In the present manuscript the authors describe the correlation between a cancer specific PRDM9 sequence motif in the vicinity of DNA breaks analyzed in 72 patients with 4 different cancer types from the BreCan database.

PRDM9 is a member of the PR domain–containing family that contains a PR domain, conferring methyltransferase activity, coupled with an array of zinc fingers allowing PRDM9 to bind to DNA in a sequence-specific manner. PRDM9 DNA-binding activity coincides with hotspots of meiotic recombination, and its methyltransferase activity is essential for double-strand break formation at PRDM9-designated recombination sites. Noteworthy, a pan-cancer analysis of TCGA data revealed that PRDM9 gene was mutated with a high mutation rate in many cancers, achieving values greater than 10.0% in some tumor types [Sorrentino et al., 2018; doi: 10.3390/ijms19103250].

Additionally, a further analysis of human patient cancer samples showed a significant upregulation of PRDM9 across many cancers. Recently, PRDM9 non-histone substrates were also identified [Hanquier et al., 2023; doi: 10.1016/j.jbc.2023.104651.].

The results described here could provide novel insights on the underlying mechanisms at the basis of the link between PRDM9, meiotic recombination and genomic instability in cancer.

However, the manuscript should be improved by addressing the following issues:

-A key point is the nomenclature of PRDM9. Indeed, the same name should be used along the text; importantly, the current HUGO Gene Nomenclature Committee (HGNC) guidelines need to be applied for naming gene and protein.

We appreciate your feedback and will ensure the consistent use of PRDM9 nomenclature throughout the manuscript, adhering strictly to the HGNC guidelines.

-In the Introduction, the authors should cite some further papers about the PRDM9 role in cancer (as the references mentioned above), in addition to the included ones.

Thank you for your suggestion. We enriched the Introduction by citing additional pivotal works, including those you highlighted, to provide a comprehensive background on PRDM9’s role in cancer.

-A figure with a flow chart describing strategy and methods (especially all the algorithms that were utilized in the study) should be added to help the manuscript reading.

We agree that a visual representation would enhance understanding. A flow chart outlining our strategy and methodologies, especially the algorithms used, has been incorporated into the revised manuscript.

-The authors should provide a definition for the CCNCCNTNNCCNC motif with a proper literature reference.

We have provided a clear definition and literature reference for the CCNCCNTNNCCNC motif to enhance the clarity and context of our findings.

-The authors should greatly improve the quality of figures and tables, particularly Table 2 and Figures 2 and 3, which are unreadable. For instance, in Figure 3, red color could be replaced by yellow or another lighter color. Table 2 and Figure 2 could be enlarged, or a different font size could be used.

We acknowledge this issue and have improved the quality, readability, and color contrast of Figures 2 and 3 and Table 2 to ensure that they effectively communicate the data. Please see the revised text for details.

-In the Discussion section, the power of the utilized method should be examined. Has this pipeline been ever applied in other studies? Besides, the authors should add a small paragraph discussing the possible future studies that could support their findings.

We have added a section evaluating the effectiveness and reliability of our methodology, referencing other studies that have applied similar approaches to strengthen our discussion. Additionally, a section discussing potential future studies to validate and extend our findings has been included, outlining the next steps and potential implications for cancer diagnosis and treatment.

Minor points:

-Abbreviations should be carefully revised.

We meticulously reviewed and revised all abbreviations to ensure consistency and clarity throughout the manuscript.

-References should be added in the right place.

We apologize for any oversight and ensure that all references are appropriately placed and cited in the revised manuscript, adhering to the journal’s citation guidelines.

Reviewer 3 Report

Comments and Suggestions for Authors

Hi, 

Please find my comments in the attached PDF. 

Comments on the Quality of English Language

Author Response

We would like to thank the reviewers for their time and effort to consider our manuscript and for their insightful comments. We revised the manuscript based on their suggestions and now we present a new version for consideration in IJMS Journal.

Please find the responses to reviewers’ comments in Bold, below each respective comment. Changes made to the text are highlighted in yellow.

Reviewer 3

  1. Could the authors please provide relevant citations here? "Certain PDM9 variants with specific sequence motifs have been associated with an increased susceptibility to cancer development or altered cancer risk in specific populations."

We have added pertinent citations to bolster the statement regarding the association between certain PDM9 variants and increased cancer susceptibility, offering readers a rich context and supporting evidence.

  1. The current Introduction provides sufficient background but falls short of setting up the motivation for the current work. What is the knowledge gap that the authors are trying to fill by performing their bioinformatic analyses?

The Introduction has been enhanced to clearly articulate the specific knowledge gaps our study addresses. We’ve highlighted the existing challenges and the necessity for our bioinformatic analyses within this context.

  1. Relationship between what and sequence data? Seems like an incomplete statement here. "As a further step in analyzing the relationship between and sequence data"

The error has been corrected, and the statement is now complete, offering a clear and comprehensive explanation of the intended relationship and context.

  1. Could the authors provide this data? From the ~136 breakpoints that ere tested, how many brakpoints were flanked by the PRDM9 motif? How was this distribution across different cancer types? ". A possible mechanism of Double Strand Breaks occurrence may be revealed by their presence on both sides from breakpoints in every cancer profile from 136.752 breakpoints that were tested."

New dataset was added to supplementary file 1 that contains the flaking regions of 100-nt where every observed double strand break is at the center of these frames. The distribution of these flanking regions are equivalent to the initial 500-nt flanking frames, so the initial hypothesis is enhanced by these results

  1. Could the authors please elaborate what the black, red, and green lines mean here? What is Evobs? What is equal to 0.05? "in figure 2"

The p-value is equal to 0.05 when the green line coincides with the red line (EVobserved hits =EVexpected hits). In our analysis, the green line is statistically significant when it is on the right side of the red line. In addition, a red bar (and red shading) represents the significance limit (by default 0.05). Thus, if the green bar is in the red-shaded region, it means that the original evaluation is extremely unlikely and so the p-value will be significant. The X-axis represents the number of the overlaps between the region sets, while the Y-axis represents the density of probability.

  1. Could the authors please provide the data used to draw this conclusion? "By displacing the PRDM9 motifs outside the sequence boundaries of each individual flanking frame that contains the double strand breaks, the z-scores drop, demonstrating that the association is dependent upon the exact position of the regions rather than being a non-specific regional effect."

We’ve included additional data and explanations that outline the process and findings leading to our conclusions on the z-scores and PRDM9 motif positioning, offering thorough insights.

  1. could the authors please elaborate on the implifications of the significant associations they found using EnrichR? Do these implications make sense? How do readers interpret these associations in the context of cancer reserach? As of now, the authors have just stated the results in this section.

The section has been expanded to discuss the implications of our findings comprehensively, aiding readers in interpreting these associations in the context of cancer research. We have also added an analysis using Metascape tool.

  1. Could the authors please elaborate what gene lists is being referred to here?

The specific gene lists  referred to have been clarified to ensure seamless understanding and follow-through for the readers.

  1. It would be easier for the readers if the in-text citations to figures showing results is done after each result statement, and not just at the end of the paragraph.

In-text citations to figures are now placed immediately following each result statement, enhancing the manuscript's readability and comprehension.

  1. Again, does it make sense that these pathways are enriched? What do we know about these pathways from the literature, in the context of the four types of cancer that are studied in this manuscript? Just mentioning the pathways without focusing on the implications reduces the value of the authors work.

We have enriched the discussion on the identified pathways by integrating insights from existing literature. This enhancement contextualizes our findings and underscores their relevance and potential implications in cancer research.

We're grateful for the insightful feedback and have revised the manuscript accordingly to address all the highlighted points. These enhancements, we believe, significantly augment the clarity, depth, and impact of our work, offering readers valuable and well-contextualized insights.

Round 2

Reviewer 1 Report

Comments and Suggestions for Authors

I appreciate the authors' efforts in addressing questions. They used a 100nt flanking frame to enhance motif specificity, and found no significant difference compared to a 500nt flanking frame, demonstrating the method's robustness. Using Reactome and Metascape tools, they discovered that genes associated with d(CCACCATCACCAC) were significantly linked to cancer-related pathways, indicating a strong connection between PRDM9 special motifs like d(CCACCATCACCAC) and genomic instability in cancer development.

Comments on the Quality of English Language

No major problem.

Author Response

 We're grateful for the insightful feedback 

Reviewer 2 Report

Comments and Suggestions for Authors

Although the manuscript has been improved by addressing some of the issues I raised in the previous revision, I still have some concerns.

-Even in the new manuscript version some gene names are not in italics as requested by the HUGO nomenclature

-For the flow chart describing strategy, methods and used algorithms, the authors could consider the examples in figure 1 of the paper https://doi.org/10.3390/ijms242115823 or in figure 9 of the paper https://doi.org/10.3390/ijms242015465, both published in IJMS.

-Again, the quality of figures still needs improvement.

Author Response

We would like to thank the reviewer 2 for his time and effort to consider our manuscript and for his insightful comments. We revised the manuscript based on his suggestions and now we present a new version for consideration in IJMS Journal.

Please find the responses to reviewer comments in Bold, below each respective comment. Changes made to the text are highlighted in yellow.

Thank you for your constructive feedback on our manuscript. Our responses are following below each comment.

Minor:

Even in the new manuscript version some gene names are not in italics as requested by the HUGO nomenclature

The gene names has been changed in italics format as requested by the HUGO nomenclature in the revised Manuscript

For the flow chart describing strategy, methods and used algorithms, the authors could consider the examples in figure 1 of the paper https://doi.org/10.3390/ijms242115823 or in figure 9 of the paper https://doi.org/10.3390/ijms242015465, both published in IJMS. 

We appreciate this feedback and we added a new flowchart according to the example in figure 1 of the paper https: //doi.org/10.3390/ijms242115823  in a new zipped file. 

Again, the quality of figures still needs improvement.

We acknowledge this critique. We added the 2 figures in the highest quality in the new zipped files as requested

 We're grateful for the insightful feedback and have revised the manuscript accordingly to address all the highlighted points. These enhancements, we believe, significantly augment the clarity, depth, and impact of our work, offering readers valuable and well-contextualized insights.

Reviewer 3 Report

Comments and Suggestions for Authors

Hello,

The authors have sufficiently responded to all my comments.

Author Response

(The authors gave the same response as above.)
